## [Reviewer comments · BMJ Open]

ARTICLE DETAILS

TITLE (PROVISIONAL)	Pharmacist-led adherence support in general practice: a qualitative interview study of adults with asthma
AUTHORS	Mes, Marissa; Katzer, Caroline; Wileman, Vari; Chan, Amy; Horne, Robert; Taylor, Stephanie

VERSION 1 – REVIEW

REVIEWER	Louise Deeks Pharmacist Researcher University of Canberra, Australia
REVIEW RETURNED	26-Jun-2019

GENERAL COMMENTS	Thank you for giving me an opportunity to review this manuscript. I have some suggestions that I believe will improve your manuscript. The research is timely and will be valuable to inform the development of clinical pharmacy services for chronic conditions in general practice. Well done! The reviewer also provided a marked copy with additional comments. Please contact the publisher for full details.
---

REVIEWER	Associate Professor Betty Char School of Pharmacy- Faculty of Medicine and Health The University of Sydney Australia
REVIEW RETURNED	05-Jul-2019

GENERAL COMMENTS	Thank you for the invitation to review this manuscript. It is well written, as can be expected. I have put forward some suggestions or reflections that could improve the paper in general. The premise upon which the method for this study was determined was, in my view, somewhat unfortunate. As mentioned in the limitations, one cannot gauge 'perceptions' of anything not yet experienced, and for people who might never experience it such as those from Scotland and N.Ireland. Perception can be interpreted as insight, and that can really only be tested or probed after experience. Perhaps the choice of words could be reviewed? e.g. opinions/expectations of patients with asthma about acceptability of a new service provided by pharmacists? If this service has been rolled out already in the UK, as mentioned in the manuscript, then the study may have been more beneficial to recruit those who have experienced the service. On the other hand I can see there is merit in asking opinions of stakeholders about what they anticipate, but
--

	that would have to be before implementation. The service has been implemented. Time for evaluation and perceptions of those who have experienced or who have clearly objected to the service. Results of this study point to barriers that should have been highlighted before implementation. I also note the sample size is not adequate - noted also by the authors, but the reason is not really the saturation or the numbers, but the research question to start with, and the fact that the sample might not actually be regarded as representative (in what way was the sample representative? there is no explanation). To describe some participants as "White" several times in this manuscript is quite extraordinary. I am not sure one could actually establish such a descriptor over the phone. And for what purpose? Was there any analysis of any correlation between these 'perceptions' and ethnicity or colour/race? Was there any finding to correlate 'perceptions' of different age groups? Or gender? So, in retrospect these demographic dimensions were not relevant. The fact that there were in reality only 3 more female participants than males [10:7] cannot be referred to as a clear majority of 'mostly' female. Recruitment in general was no strength in this study. When reporting such numbers percentages really mean little in the scheme of things. I believe the discussion could have been deeper, albeit with all the limitations mentioned. Trust and reliability are important aspects of any service provided in pharmacy and warranted some further delving into the quotes provided; for example, tying in age and gender perhaps, or severity of the asthma they suffered. Noticeable were the two older participants who were more willing to utilise the service. Not really presented this way, but would be of interest to see what age group needs to be targeted with promotion/s in the future. Reporting of themes and subthemes were sometimes vague e.g. in the subtheme 'Potential role overlap" the examples given were those who 'clearly delineated the GPP role'. Discussion was brief. This was an opportunity to offer e.g. recommendations, analyses and merits of findings with the objective of enhancing uptake of the service - not regurgitation of results. Implications for research and practice was a better section, putting forward a reasonable discussion of future possibilities. References 2 and 4 might benefit from a link or an ISBN. Formatting, in particular the headings and subheadings appeared to need attention.
--	---

VERSION 1 – AUTHOR RESPONSE

Reviewer Evaluation: Reviewer 1

Thank you for giving me an opportunity to review this manuscript. I have some suggestions that I believe will improve your manuscript. The research is timely and will be valuable to inform the development of clinical pharmacy services for chronic conditions in general practice. Well done!

Thank you for taking the time to review our manuscript and for your encouraging feedback. We have considered your comments carefully and amended the manuscript accordingly; please see the points below for further details. We hope these amendments meet with your agreement. However, please let us know if there is anything further that is required.

- Abstract: clarified that the study 'aimed to explore the perspectives of adults with asthma on the potential for pharmacist-led adherence support...'
- Strengths and limitations of this study: clarified that participants in the sample 'primarily had self-reported mild asthma' (and this point was also added to our discussion as a limitation)
- Methods:
 - o Stated and referenced the use of the Standards for Reporting Qualitative Research (SRQR) guidelines
 - o Clarified who transcribed the interviews and who checked for accuracy
 - o Stated the start and end dates of the study
 - o Clarified that 4 transcripts (25% of all transcripts) were independently coded by another researcher (with a reference for this proportion).
- Discussion: clarified that broad clinical skills and prescribing qualifications are programme priorities in the GPP training pathways designed by the Centre for Pharmacy Postgraduate Education.

Reviewer Evaluation: Reviewer 2

Thank you for the invitation to review this manuscript. It is well written, as can be expected. I have put forward some suggestions or reflections that could improve the paper in general.

The premise upon which the method for this study was determined was, in my view, somewhat unfortunate. As mentioned in the limitations, one cannot gauge 'perceptions' of anything not yet experienced, and for people who might never experience it such as those from Scotland and N.Ireland. Perception can be interpreted as insight, and that can really only be tested or probed after experience. Perhaps the choice of words could be reviewed? e.g. opinions/expectations of patients with asthma about acceptability of a new service provided by pharmacists?

Thank you for taking the time to review our manuscript and provide comments. We agree that the use of 'perceptions' suggests that participants experienced the pharmacist-led services first-hand. We have amended the wording in the manuscript from 'perceptions' to 'opinions' and 'perspectives of adults with asthma on the potential of pharmacist-led adherence support'. We hope that this amendment clarifies the issue.

If this service has been rolled out already in the UK, as mentioned in the manuscript, then the study may have been more beneficial to recruit those who have experienced the service. On the other hand I can see there is merit in asking opinions of stakeholders about what they anticipate, but that would have to be before implementation. The service has been implemented. Time for evaluation and

perceptions of those who have experienced or who have clearly objected to the service. Results of this study point to barriers that should have been highlighted before implementation.

We agree with this comment and recognise participants' lack of experience of pharmacist-led consultations as a study limitation in the discussion. We believe that the study findings can still have a positive impact on clinical practice, as the general practice pharmacist scheme in England is still in its early stages (with the aim of recruiting 2,000 clinical pharmacists to general practice by 2020). Furthermore, general practice pharmacists are yet to be introduced in Scotland, Wales and Northern Ireland. As the scheme is rolled out, findings from this study can be used to improve service uptake among adults with asthma and also inform service delivery in other countries.

Dr. Claire Mann at the School of Pharmacy at the University of Nottingham is leading the evaluation of the general practice pharmacist scheme in England, including interviews with service users (patients).¹ The initial evaluation of the general practice pharmacist pilot scheme took a broad approach across several long-term conditions and we felt that we could contribute to existing efforts by offering asthma-specific insight, as this level of detail was not included in the published report.

The focus on asthma is particularly timely and important as recent research shows that engagement with basic asthma care is low among people with asthma in the UK.² Therefore, research with service users (patients already engaged in their care) may not be able to provide important insight on how to improve service uptake and overall engagement with new healthcare services among people with asthma.

I also note the sample size is not adequate - noted also by the authors, but the reason is not really the saturation or the numbers, but the research question to start with, and the fact that the sample might not actually be regarded as representative (in what way was the sample representative? there is no explanation).

We agree with your comment and have recognised the sample size as a limitation in our discussion. Many of our participants were recruited through Asthma UK (41%) and these individuals may have been more engaged in asthma care compared to the general population with asthma and therefore may not be able to be regarded as representative. However, we believe that the doubts about pharmacist-led services expressed by study participants who were more engaged in asthma care may be amplified in the general population with asthma with less of an interest in asthma care, as outlined in the discussion.

To describe some participants as "White" several times in this manuscript is quite extraordinary. I am not sure one could actually establish such a descriptor over the phone. And for what purpose? Was there any analysis of any correlation between these 'perceptions' and ethnicity or colour/race? Was there any finding to correlate 'perceptions' of different age groups? Or gender? So, in retrospect these demographic dimensions were not relevant.

Thank you for your comment and we apologise for the confusion. As stated in the 'data collection and analysis' section, we collected demographic details (including self-reported ethnicity) through an online questionnaire as we agree that this would be inappropriate to establish over the telephone. We have amended the manuscript to try to clarify this even further.

We initially reported all demographic details to give the reader an overview of the participant sample. However, we do agree that we should have commented on potential links between these characteristics and people's opinions of general practice pharmacists, and we have expanded on this point in the 'interpretation of findings' section.

As we did not establish any notable differences in opinions based on demographic characteristics, we have removed them from the manuscript and kept only age and gender, as is common practice in BMJ Open publications and as is stated in the Standards for Reporting Qualitative Research (SRQR) checklist. We hope that these amendments help clarify this point and meet with your agreement.

The fact that there were in reality only 3 more female participants than males [10:7] cannot be referred to as a clear majority of 'mostly' female. Recruitment in general was no strength in this study. When reporting such numbers percentages really mean little in the scheme of things.

We agree with this comment and have amended the manuscript accordingly, both in the results and discussion sections.

I believe the discussion could have been deeper, albeit with all the limitations mentioned. Trust and reliability are important aspects of any service provided in pharmacy and warranted some further delving into the quotes provided; for example, tying in age and gender perhaps, or severity of the asthma they suffered. Noticeable were the two older participants who were more willing to utilise the service. Not really presented this way, but would be of interest to see what age group needs to be targeted with promotion/s in the future.

Discussion was brief. This was an opportunity to offer e.g. recommendations, analyses and merits of findings with the objective of enhancing uptake of the service - not regurgitation of results.

Thank you for these comments. In line with your feedback, we have replaced the 'main findings' section of the manuscript with an 'interpretation of findings' section that discusses the importance of trust in patient-pharmacist relationships, and the link between participants' gender/age/self-reported asthma severity and their views of general practice pharmacist-led care. We also discuss ways to increase trust in pharmacists based on our findings regarding the hierarchy of healthcare professionals and concerns raised by participants about pharmacist-led care. We hope this revised section offers a deeper discussion of our findings, rather than a summary of the main results.

We agree that the willingness of the two older participants to use pharmacist-led services was a notable finding, and we have highlighted it under 'comparison with existing literature'.

Reporting of themes and subthemes were sometimes vague e.g. in the subtheme 'Potential role overlap' the examples given were those who 'clearly delineated the GPP role'.

Thank you for your comment. We thought it important to highlight both people who did and did not delineate between pharmacists and other healthcare professionals. Based on your feedback, we have re-structured this section to clarify that we are offering both perspectives and providing supporting quotes to illustrate both views.

Implications for research and practice was a better section, putting forward a reasonable discussion of future possibilities.

References 2 and 4 might benefit from a link or an ISBN.

Formatting, in particular the headings and subheadings appeared to need attention.

Thank you for these encouraging comments. We have added links to the references as suggested and reviewed the formatting of our headings and subheadings in line with BMJ Open formatting.

1. Mann C, Anderson C, Avery AJ, et al. Clinical pharmacists in general practice: pilot scheme. Independent Evaluation Report. University of Nottingham, UK: NHS England, 2018.
2. Cumella A. Falling through the gaps: Why more people need basic asthma care. Annual Asthma Survey. United Kingdom: Asthma UK, 2017.

VERSION 2 – REVIEW

REVIEWER	Louise Deeks Discipline of Pharmacy University of Canberra Australia
REVIEW RETURNED	10-Sep-2019

GENERAL COMMENTS	Thanks for giving me the opportunity to review your revised manuscript. Most of the comments from the previous review have been addressed – well done. I have attached some additional comments which I believe will improve your manuscript.
--

VERSION 2 – AUTHOR RESPONSE

Reviewer Comments

Thanks for giving me the opportunity to review your revised manuscript.

Most of the comments from the previous review have been addressed – well done.

I have the following comments:

Strengths and limitations of this study:

Point one – consider changing perceptions to opinions/perspective

Results:

Did the interview lengths have a normal distribution? If not, suggest median (and range) better than mean (and standard deviation)

Theme 3: The hierarchy of healthcare professionals.

Paragraph 2: consider changing perceptions to opinions/perspective

Discussion:

Paragraph 4: However, participants on multiple medications or those with additional health concerns

I am unclear how data on the number of medications or other health conditions of participants was collected. Please clarify.

Comparison with existing literature:

Paragraph 1:

However, a few of the older participants in this study were open to GPP-led care

Can you specify a number rather than use 'a few'?

Paragraph 2:

Please link to your research

Both general members of the public and people with asthma have been found to use other trusted healthcare professionals (e.g. GPs and nurses) as benchmarks for trust when asked about a community pharmacist-led service.(13 20) Similarly, Naik Panvelkar, et al. 20 found that previous positive experiences with community pharmacists raised expectations for other pharmacist-led services in a population of people with asthma.

References

Suggest use doi: consistently - remove <https://dx.doi.org/> where used

- add for all journal references

For internet references – please add date accessed

Ref 22 & 24 – journal title is missing

Reporting checklist:

The page numbers do not match the reporting items. Please rectify.